# Early Detection of Respiratory Diseases in Calves by Use of an Ear-Attached Accelerometer

**DOI:** 10.3390/ani12091093

**Published:** 2022-04-23

**Authors:** Nasrin Ramezani Gardaloud, Christian Guse, Laura Lidauer, Alexandra Steininger, Florian Kickinger, Manfred Öhlschuster, Wolfgang Auer, Michael Iwersen, Marc Drillich, Daniela Klein-Jöbstl

**Affiliations:** 1Clinical Unit for Herd Health Management in Ruminants, University Clinic for Ruminants, University of Veterinary Medicine, 1210 Vienna, Austria; nasrin.ramezani@vetmeduni.ac.at (N.R.G.); christian.guse@vetmeduni.ac.at (C.G.); michael.iwersen@vetmeduni.ac.at (M.I.); marc.drillich@vetmeduni.ac.at (M.D.); 2Smartbow GmbH/Zoetis LLC, Jutogasse 3, 4675 Weibern, Austria; lauralidauer24@gmail.com (L.L.); alexandra.steininger@zoetis.com (A.S.); florian.kickinger@zoetis.com (F.K.); manfred.oehlschuster@zoetis.com (M.Ö.); wolfgang.auer@aisemo.com (W.A.)

**Keywords:** bovine respiratory disease, calf behavior, lying behavior, accelerometer

## Abstract

**Simple Summary:**

Bovine respiratory disease is one of the most important diseases in group-housed calves worldwide, with impacts on calf welfare and farm economics. Early detection of the disease is important for the well-being of the animals and a targeted treatment. Therefore, tools for an automated monitoring of individual calves would be a breakthrough in health management. In this study, we used an ear-attached accelerometer to evaluate its potential for the early detection of behavioral changes related to respiratory disease in calves. Our result showed that accelerometers are able to detect changes in activity and lying times that can be used to predict respiratory disease before clinical diagnosis.

**Abstract:**

Accelerometers (ACL) can identify behavioral and activity changes in calves. In the present study, we examined the association between bovine respiratory disease (BRD) and behavioral changes detected by an ear-tag based ACL system in weaned dairy calves. Accelerometer data were analyzed from 7 d before to 1 d after clinical diagnosis of BRD. All calves in the study (n = 508) were checked daily by an adapted University of Wisconsin Calf Scoring System. Calves with a score ≥ 4 and fever for at least two consecutive days were categorized as diseased (DIS). The day of clinical diagnosis of BRD was defined as d 0. The data analysis showed a significant difference in high active times between DIS and healthy control calves (CON), with CON showing more high active times on every day, except d −3. Diseased calves showed significantly more inactive times on d −4, −2, and 0, as well as longer lying times on d −5, −2, and +1. These results indicate the potential of the ACL to detect BRD prior to a clinical diagnosis in group-housed calves. Furthermore, in this study, we described the ‘normal’ behavior in 428 clinically healthy weaned dairy calves obtained by the ACL system.

## 1. Introduction

Bovine respiratory disease (BRD) is one of the most important problems in dairy cattle older than 30 days of age [1,2]. It is a multifactorial disease caused by various pathogens [3] and influenced by the interaction between pathogens, immunity of animals, environment, and management [4]. Mortality associated with BRD in calves varies between 9.4 and 22.5% [4,5,6]. Impaired welfare [7], reduced growth rates [8], and increased treatment costs [9] are the most important consequences of BRD. The typical clinical signs of BRD are depression, anorexia, increased respiratory rate, as well as nasal and ocular discharge, cough, and fever [10,11]. These signs, however, are not specific for respiratory disorders. Detection of BRD is often based on physical examination of individual animals. This is time consuming and often not feasible in larger groups of animals. Hence, clinical signs-based scoring systems in groups of animals were introduced [12]. New technologies for an automated identification of calves suffering from BRD could improve animal welfare monitoring. 

In the last few years, accelerometers became an essential part of ‘precision livestock farming’ (PLF) representing a reliable and affordable technology to monitor feeding and behavioral changes and to manage farm animals performance [13]. These technologies are able to measure and record activity, lying and rumination times to detect changes in behavior that might be indicative of diseases and other conditions, e.g., estrus [14,15]. Many recent studies showed the high potential of accelerometers to monitor rumination, activity, and lying behavior in calves [16,17,18]. It has been reported that these behavioral changes can be detected in diseased calves before clinical signs are evident [19]. The objective of this study was to determine the difference between BRD cases and clinically healthy controls in behavior using an ear-attached accelerometer (ACL) in order to identify affected calves prior to the clinical detection of the disease. Furthermore, we aimed to describe normal behavior by use of the ACL system, as there are no studies that specifically describe daily and weekly activity, lying, and rumination patterns in weaned group-housed dairy calves.

## 2. Materials and Methods

### 2.1. Animals, Experimental Design and Housing

The study procedures were approved by the Slovakian Regional Veterinary Food Administration and noted by the institutional ethics and animal welfare committee of the University of Veterinary Medicine, Vienna (ETK-11/09/2017).

The study was conducted on a commercial dairy farm in Slovakia between January and November 2019. The study farm housed approximately 2700 Holstein dairy cows and young stock. On farm, all calves were housed individually in hutches for the first two months of life (until weaning). Afterwards, they were housed in groups of five, later 10 animals, and at the age of four months they were moved to another barn on the same farm, where the present study took place. The calves under study were housed in barns with groups of approximately 30 animals in 7.7 × 13.5 m pens with concrete floors, covered with straw. Manure and soiled bedding were removed every week and replaced by clean straw. Calves had free access to water and feed. A total mixed ration (TMR) was fed to the calves twice a day. Based on the dry matter, the TMR consisted of 33% corn silage, 17% grass silage, 16% concentrates, 9% wet distillers grain with soluble, 8.5% soy extraction meal, 6% corn cob mix, 5% straw, 4% soy extraction meal, and 1.5% minerals. On farm, all female calves were vaccinated against *Bovine Parainfluenza Virus type* 3, *Bovine Respiratory Syncytial Virus* (Rispoval, Zoetis, Louvain-la-Neuve, Belgium), *Bovine Herpesvirus type 1* (Bovilis IBR Marker live, MSD Animal Health, Milton Keynes, United Kingdom), and *Moraxella bovis* (MORAXEBIN Neo, Bioveta a.s., Czech) within the first weeks of life.

### 2.2. Accelerometer Data Collection 

A 10 Hz accelerometer-equipped ear-tag (SMARTBOW, Smartbow GmbH/Zoetis LLC, Weibern, Austria) with a size of 52 × 36 × 17 mm and a weight of 34 g was attached to the left ear. All calves (n = 508) entering the study barn were equipped with the ear-tag approximately 2–3 weeks before potential enrollment. Data were wirelessly transmitted from the ear-tag to the receivers (Smartbow wallpoints) every 4 s if a calf was active and every 16 s if a calf was inactive. The continuously recorded acceleration data (raw data) were further processed by proprietary algorithms, originally developed by the manufacturer for adult cows, on a farm server [20,21,22]. Classified data based on these algorithms for lying, standing, active, inactive, high active, and rumination were presented visually on a local computer or on a mobile device and recorded in the SMARTBOW software. All parameters were presented as minutes per hour (min/h) that the animal spent with this activity. As animals could either be active, inactive, or high active, the sum of these three parameters was always 60 min/h.

### 2.3. Clinical Score Assessment and Definitions 

The 508 clinically healthy calves without fever (fever = rectal temperature ≥ 39.5 °C) and specific clinical signs of BRD, and not being treated with antibiotics before entering and during the study, were eligible for the study and scored daily by one veterinarian (first author, N.R.G) according to an adopted University of Wisconsin Calf Scoring System for group-housed calves [12]. The daily scoring included evaluation of cough, nasal discharge, head and ear position, and ocular discharge. Each parameter was classified using a four-point scale, where 0 was considered normal and 3 as severely abnormal (Table 1). Scores for each parameter were recorded on preprinted data capture forms and transferred into Microsoft Excel (Excel 2010, Microsoft, Redmond, WA, USA). On each study day, all calves with a total respiratory score (summarizing all parameters to one score) of ≥4 or at least two parameters with a score ≥2 were fixed in headlocks for further clinical examination. 

The clinical examination included lung auscultation for respiratory sounds (normal, strong “F” sound (increased vesicular sound), crackles, wheezes, and absence [23]), examination for dyspnea (normal, moderate, and severe), and measurement of the rectal temperature. All calves with a rectal temperature ≥39.5 °C and at least one parameter ≥2 were defined as ‘at risk’ for BRD. Calves were only defined as diseased (DIS) if the rectal temperature ≥39.5 °C and at least one parameter ≥2 were present for two consecutive days. Rectal temperature was measured by use of a digital thermometer (Veterinär-Thermometer SC 1080, Scala Electronic GmbH, Stahnsdorf, Germany). Day 0 was defined as the day when BRD was first diagnosed, whereas d +1 was the day of confirmation. For each calf ‘at risk’ and diseased, respectively, a clinically healthy control calf (CON) of the same age, with total score number 0 or 1, was further examined. If the calf did not show any sign of disease during clinical examination it was chosen as CON. Calves ‘at risk’ that could not be classified as diseased during the clinical examination on two consecutive days were excluded from the study, but the respective control calves stayed. Consequently, the number of CON was higher than the number of DIS. In this study, the classified measures of the ACL, i.e., activity, lying, and rumination time were compared between DIS and CON 7 d before to 1 d after BRD diagnosis. 

### 2.4. Additional Examinations

All calves were weighed on the day of enrollment and two weeks later, and the average daily weight gain (ADG) was calculated. To screen for pathogens present on the farm, deep nasal swabs (EYDAM; Tupfersystem-lange Ausführung, Kiel, Germany) were taken approximately every two weeks during the study period from a DIS and CON animals, which were randomly chosen by use of the Excel random function. In total, 35 samples (DIS = 18, CON = 17) were examined at the Institute of Microbiology and the Institute of Virology at the University of Veterinary Medicine, Vienna. For microbiology, clinically relevant bacteria were isolated in initial characterization from samples by cultivation on standard agar plates. Subsequently for species identification, *Pasteurella multocida* and *Mannheimia haemolytica* were identified by Matrix-Assisted Laser Desorption/Ionization-Time of Flight Mass Spectrometry (MALDI-TOF MS). The samples were tested by RT-PCR for *Bovine Respiratory Syncytial Virus* (BRSV), *Parainfluenza Virus type 3* (PIV3), and *Bovine Corona Virus* (BCoV) specific nucleic acids. 

Following the clinical examination, DIS and CON calves (DIS = 38 and, CON = 38) were examined by thoracic ultrasonography (5-MHz linear-array transducer, Easi-Scan, BCF Technology Ltd., Bellshill, Scotland, UK) [24,25]. 

### 2.5. Normal Behavior 

During the study period, we described the normal behavior in 428 clinically healthy calves with eligible ACL data that showed no sign of BRD or other diseases. ‘Normal behavior’ is presented for a period of one week and, in more detail, for one day.

### 2.6. Statistical Analyses 

All data were transferred to Microsoft Excel (Excel 2010, Microsoft, Redmond, WA, USA). Statistical analyses were performed in R environment version 4.0.2 using the package caret [26], and SPSS (version 24, Amrok, New York, NY, USA). Before analysis using the Python package pandas (Python 3.7.9, pandas 1.0.3), only ACL data labelled as valid by SMARTBOW (i.e., at least data for 40 out of 60 min were available) were selected and summarized per calf and day starting with d −7 to d +1 relative to first diagnosis of BRD in DIS (d 0). Averages were reported as mean ± SD. Normality of all variables was assessed using the Shapiro–Wilk test. As some variables were normally distributed and others were not (see Appendix A), a Mann–Whitney U test was performed to test for an association between the duration of activity level, lying, and ruminaton of DIS and CON calves. *p*-value < 0.05 was considered as significant.

To evaluate the adequacy of ACL data to predict the calves’ health status (in DIS and CON) on subsequent days (d −7 to d −1) separately, Lasso regularized multivariate logistic regression models (L1 RMLM) were used for a selection of multiple days from the dataset of CON and DIS for the final model. These models allow evaluation of multiple or continuous predictors and estimation of reliable predictor coefficients, even when predictors are highly correlated or the sample size is small [27]. The models included the calves’ health status, behavioral parameters (active, inactive, high active, lying, rumination time, and their respective interaction terms), age, and season (as classifier; 1 = spring, 2 = summer, 3 = autumn, 4 = winter). The data from d −7 to d −1 were standardized to z-scores. Regularization parameters were determined using three times 10-fold cross validation. Pre-processing of the data included scaling and centering.

The models ability to detect sick calves were described by area under the curve (AUC), accuracy (ACC), sensitivity (SE), specificity (SP), positive predictive values (PPV), and negative predictive values (NPV). The two best models based on ACC and AUC were selected for a combined L1 RMLM with the dataset then randomly split into 70% training and 30% test, while maintaining the distribution of health status.

## 3. Results

### 3.1. Animals and Clinical Examination

In total, we examined 508 calves for respiratory disease. Mean age of these calves was 130.4 ± 13.4 d. The daily scoring of calves in groups revealed that 298 calves (58.6%) had a total score of ≥4 and were considered for further clinical examination. Of these calves, 83 (16.3%) were classified as ‘at risk’ and consequently clinically examined for 2 consecutive days. In the end, 48 calves (9.4%) were categorized as DIS. 

### 3.2. Acceleration Data 

Seven DIS calves were excluded because of data loss by the ACL system. The final number of DIS calves to compare acceleration data with CON (n = 69) was 41. The mean age of DIS was 131.7 ± 13.3 d and of CON 130.4 ± 12.2 d (*p* = 0.93). 

Activity (i.e., active, inactive, high active), lying, and rumination time for DIS and CON for the last 7 d before and up to one day after clinical diagnosis are presented in Figure 1. Generally, DIS showed more inactive times compared to CON. The differences were significant on day −4 (17.1 ± 4.0 vs. 15.8 ± 3.4 min/h, *p* = 0.049), day −2 (18.4 ± 5.5 vs. 16.7 ± 5.3 min/h, *p* = 0.04), and day 0 (17.5 ± 5.5 vs. 15.4 ± 3.9 min/h, *p* = 0.03). For more details to see Appendix A. 

Consequently, high active times were shorter in DIS compared to CON. High active values ranged between 0 and 23.0 min/h in DIS and 4.1 to 31.1 min/h in CON. Differences between groups were significant on all evaluation days, except on d −3.

Lying times were higher in DIS than in CON, whereat differences were only significant on d −5, d −2, and d +1. Active and rumination times did not differ significantly between groups.

### 3.3. Prediction Models 

The results of the L1 RMLM showed the potential for detecting BRD calves before the onset of clinical signs (Table 2). The AUC/ACC varied between a minimum on day −7 (AUC: 0.57, ACC: 0.68) and a maximum on d −2 (AUC: 0.75), and on d −3 (ACC: 0.79). The most promising model results based on AUC/ACC were seen on d −2 and d −3, therefore they were selected for a combined L1 RMLM. The performance in an out-of-sample test is noted by a SE of 71.4%, SP of 95.2%, ACC of 85.7%, and AUC of 83.0% (Table 2). Variables with coefficients different from zero included: on d −2, age, season, rumination and interaction between lying and high active, inactive and high active times and on d −3, age, season, and interaction between rumination and inactive, active, and high active time. 

### 3.4. Additional Examinations

The ADG did not differ between groups with 1.2 ± 2.5 kg/day for DIS and 1.2 ± 1.1 kg/day for CON. Results of the ultrasonographic examination in DIS and CON were obtained from 76 calves (Table 3). Three DIS and 3 CON calves could not be examined by ultrasonography due to technical problems. In 81.5% (31/38) of DIS, abnormalities (lung surfaces with multiple comet tail artifacts, abscess, lesion, lung consolidation, and pleural effusion) were found. In CON, 78.9% (30/38) showed no abnormalities; comet tails and a lesion <1 cm was evident in six and two animals, respectively. Pathogens detected in nasal swabs of 18 DIS and 17 CON calves under study were *Bovine Corona Virus* (4 DIS and 3 CON), *Pasteurella multocida* (1 DIS), and *Mannheimia haemolytica* (1 DIS). 

### 3.5. Normal Behavior

The 428 clinically healthy calves that were used to describe daily normal behavior were at a mean age of 130.8 days (SD ± 13.1 d). The mean ± SD time (min/h) of activity, (i.e., active, inactive, and high active), rumination, and lying over the period of one week and 24 h (min/h) are presented in Figure 2 and Figure 3, respectively. These animals had daily lying times of 31.0 ± 9.4, rumination times of 19.5 ± 2.9, inactive times of 16.9 ± 5.4, active times of 39.5 ± 3.7, and high active times of 3.6 ± 1.8 min/h.

Weekly means ± SD for inactive, active, and high active times were 16.9 ± 5.4, 39.4 ± 3.7, and 3.6 ± 1.8 min/h. Within one day, inactive times ranged between 7.2 ± 6.0 and 27.4 ± 7.7 (at 9:00 and 4:00 a.m.), for active between 31.8 ± 7.4 and 45.4 ± 6.6 (at 4:00 and 9:00 a.m.), and for high active between 0.8 ± 1.6 and 7.4 ± 5.1 (at 5:00 and 9:00 a.m.) min/h. The longest active times were recorded at 9:00 a.m. and 7:00 p.m. (Figure 3c).

The mean ± SD weekly lying time was 30.9 ± 9.4 min/h (Figure 2d). Daily lying times ranged from 14.4 ± 8.2 to 48.6 ± 7.3 min/h, with the longest lying times during the night (i.e., 10 p.m. to 5 a.m.; Figure 3d).

The mean value for weekly rumination was 19.5 ± 3.1 min/h. Daily times ranged between 12.8 ± 6.3 and 24.6 ± 6.7 min/h. Rumination times were longest during the night hours (1:00 to 5:00 a.m.; 24.6 ± 6.7 min/h) when simultaneously lying times were longest (48.6 ± 7.3 min/h). This result showed that daily normal rumination behavior followed periods of resting and active times in calves.

## 4. Discussion

The main objective of the present study was to test an ear-tag based accelerometer to detect behavioral changes associated with BRD for an early diagnosis. Previous studies showed that the used ACL is a reliable device to monitor rumination (correlation with visual observations, r > 0.99) [20], activity times [15], signs of estrus (SE = 97%, SP = 98%) [14], and onset of calving (SE = 54%) [16] in adult cows and lying (SE = 94%, SP = 94%) [22], drinking behavior (SE = 83%, SP = 97%) [28], and diarrhea in calves (SE = 68%, SP = 62%) [19]. 

Although the algorithm used in the study was not validated for calves, the main objective of the present study was to determine the suitability of ACL based parameters for detecting BRD before clinical symptoms appeared. Nevertheless, it has been shown that the system has the potential, e.g., to detect early indicators for newborn calf diarrhea [19]. 

Another limitation in the present study is that the number of animals classified as diseased was only 9.4%. Reported prevalences for BRD with 10–22.8% were higher in other studies [29,30,31]. The most probable explanation could be that the criteria for classifying animals as diseased in this study constituted a higher threshold for diseased status, as calves had to show signs of BRD (rectal temperature ≥ 39.5 °C and at least another parameter ≥2) for at least two consecutive days. Although only a small number of animals were examined for relevant pathogens, the results indicate a low infectious pressure. Other possible causes for a relative low number of diseased animals compared to other studies could be sufficient calves’ response to vaccination [32], differences in age of the study calves (with older calves being at higher risk [33]), and other management factors like production system and calf housing [34,35].

Depression, decreased activity, decreased feeding and rumination times, increased lying times and number of lying bouts have been associated with BRD [29,36,37,38]. It has been shown that these behavioral changes can be recognized before clinical diagnosis of disease [39,40]. We hypothesized that the ACL system might be a useful tool to detect behavioral changes, and consequently identify animals at risk of becoming diseased, early and with minimum labor and time. The results of the present study revealed that significantly decreased activity (decreased high active and increased inactive times) and increased lying times at least during selected days before disease diagnosis can be seen in DIS compared to CON calves. In contrast, no significant differences were observed for rumination. Although, not directly comparable to our study, the study of Gusterer et al. [15] with transition cows provided some evidence that rumination only changes significantly when multiple disorders are present. This could partly explain our results as well. In the literature, results regarding changes in behavioral parameters before onset of disease differ, e.g., for lying behavior. Some authors reported decreased lying bouts in calves before onset of BRD [29,39], but no significant effect on the lying times. Others found increased lying times in association with respiratory disease [41,42] or decreased lying times when calves suffered from diarrhea or navel infection [43].

The combined model revealed that the best prediction for disease can be made based on the combination of data from d −2 and d −3. This may be enough time to take advantage of an early detection by the system and to initiate an adequate treatment. The PPV and NPV showed that 90.9% of DIS and 83.3% of CON calves have been recognized correctly. Nevertheless, sensitivity (71.4%) and accuracy (85.7%) were non-satisfying. The lack of sensitivity can also be seen in similar studies with even worse results. Belaid et al. [29], for example, used an activity measuring device to detect sickness in veal calves early with the best prediction model one day before onset of disease with a SE of 68.8%, SP of 72.4%, and accuracy of 71.5%. Knauer et al. [44] reported a sensitivity of up to 74.9% (and SP of 27.1%) for the prediction of disease using either single or combined feeding behaviors. A combination of more and inclusion of additional behavioral factors (e.g., feeding behaviors, standing still, resting, and postures), respectively, may improve prediction models. Besides this, the development and validation of calf-specific algorithms for classifying animal behavior should be the scope of future research.

Data regarding ‘normal’ behavior obtained by sensor technologies in calves are scarce. To the best knowledge of the authors, this is the first time that a description of daily normal behavior has included algorithm based activity patterns (i.e., active, inactive, and high active), lying, and rumination in group-housed dairy calves at the age of 4–6 mo. Normal activity and behavior of group-housed calves may change with age and farm management [45]. Therefore, it is necessary to describe normal behavior in different age groups under different management conditions [29,46]. The mean ‘normal’ lying time in this study in clinically healthy calves was 31.0 ± 9.4 min/h (resulting in 12.4 h/day), which is lower than in other studies in younger calves (neonates and 30 to 90 d old, from 16.8 to 17.3 h/day [46] and 16.8 h/day [29], respectively). Difference between daily lying times could be due to the effect of climate [47], bedding material [48,49], and regrouping [50]. Although studies are scarce regarding the influence of age on lying behavior, such an influence could be seen with either an decrease or increase in pre-weaned and weaned calves [51]. Furthermore, it is possible that the lying times have been influenced negatively by daily examination procedures including chasing and fixing of animals in headlocks during clinical examination and ultrasonography if applicable.

We observed that clinically healthy calves ruminated for 19.5 ± 2.9 min/h (resulting in 467.0 min/day), which is similar to the value reported by Rodrigues et al. [52] (457 min/day), but longer than in the study by Lopreiato et al. [53] (373 min/day). This difference could mainly be explained by a feeding effect, as in the study of Lopreiato et al. [53] pre-weaned calves were examined [54]. This shows that ‘normal’ behavior has to be defined for different groups of animals as it is influenced by different factors (e.g., age, climate, housing, bedding, and feeding).

The findings can improve our understanding of behavioral changes and could be useful to recognize calves at risk for BRD before onset of disease to further develop a calf-specific alert system. Further studies especially with greater numbers of diseased animals are necessary to elucidate this and to confirm the results presented here. 

## 5. Conclusions

The results of this study indicate that ear-tag based accelerometers for monitoring lying and activity parameters have the potential to detect animals with BRD at an early stage. The data suggest that the use of high active and inactive times could be useful for identifying diseased calves, but further research is needed to confirm these results with a larger sample size. Sensor technology can be a reliable method for early recognition, especially for group-housed animals with lesser possibility of individual monitoring, which can guide management decisions. The development of specific algorithms for describing calf behavior could be the subject of future studies. 

## Figures and Tables

**Figure 1 animals-12-01093-f001:**
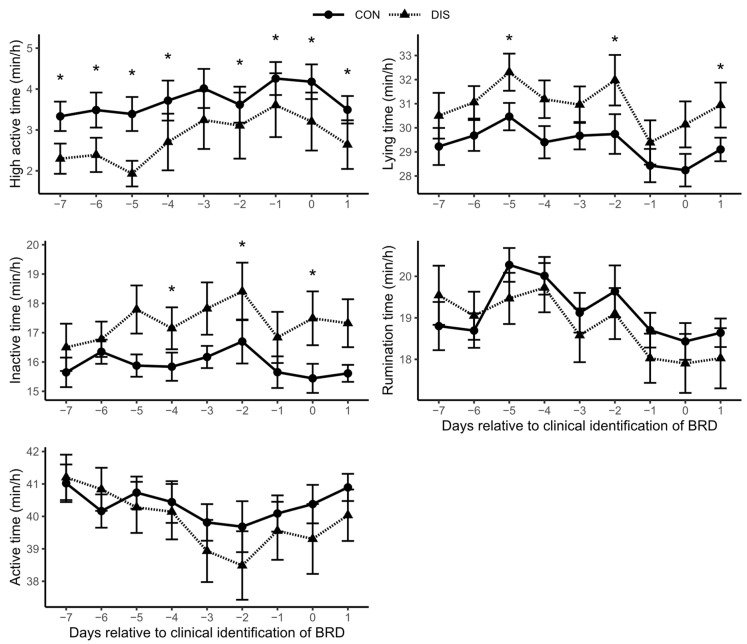
High active, inactive, active, lying and rumination times for diseased calves (DIS = 41) and clinically healthy controls (CON = 69) from d −7 to d +1 relative to clinical diagnosis of bovine respiratory disease (BRD) (d 0). * changes between groups with a *p* < 0.05 based on Mann–Whitney U test. Error bars represent standard error of the mean (SEM).

**Figure 2 animals-12-01093-f002:**
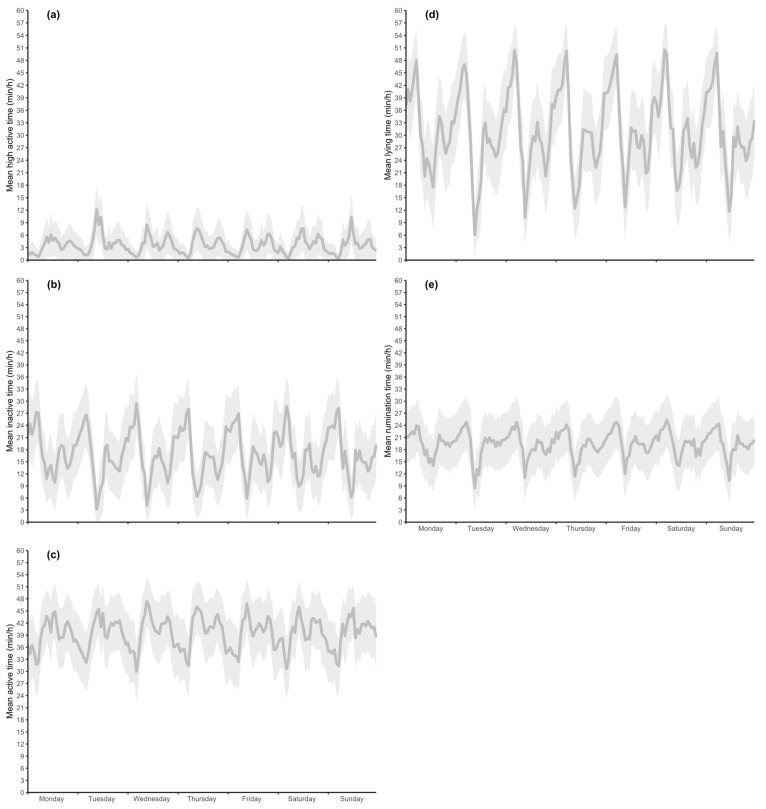
‘Normal’ behavior examined by an ear-tag based accelerometer in 428 clinically healthy calves over one week. Mean (dark grey line) and SD (light gray areas) in minutes per hour of the parameters high active (**a**), inactive (**b**), active (**c**), lying (**d**), and rumination (**e**) times are presented.

**Figure 3 animals-12-01093-f003:**
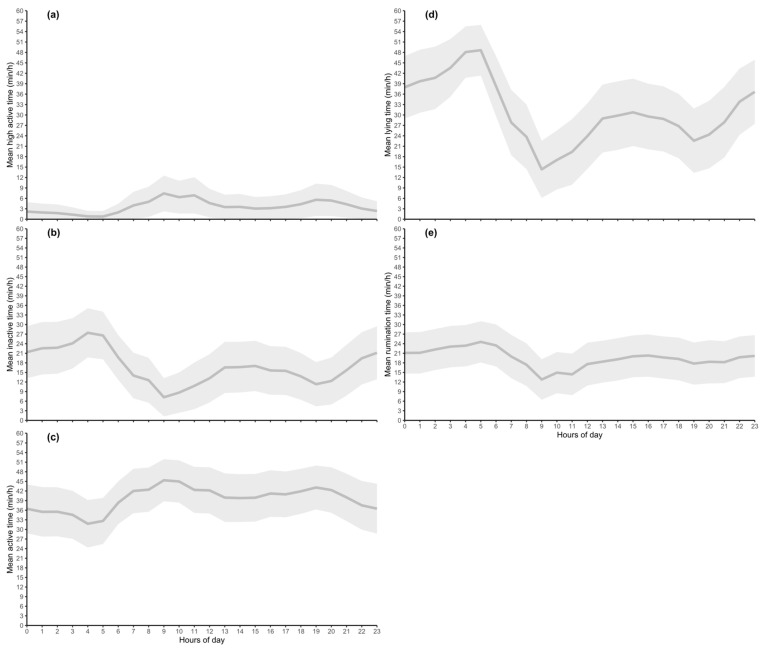
Mean ‘normal’ behavior (dark grey line) and SD (light gray area) in 428 clinically healthy calves within one day. The parameters high active (**a**), inactive (**b**), active (**c**), lying (**d**), and rumination (**e**) are presented.

**Table 1 animals-12-01093-t001:** Scoring system and definitions for daily observations in group housed calves adapted * from McGuirk and Peek [12].

Parameter	Score			
	0	1	2	3
Rectal temperature (°C)	37.5–38.2	38.3–38.8	38.9–39.4	≥39.5
Cough	none	induced-single	induced and spontaneous single (2–3)	spontaneous repeated (>3)
Nasal discharge	serous	small amount, unilateral, cloudy	bilateral, mucus	bilateral, mucopurulent
Head and ear position	normal	ear flick/head shake	slight unilateral ear drop	severe head tilt, or bilateral ear drop
Ocular discharge	none	small	moderate	severe

* Definition of scores for cough and nasal discharge were further specified.

**Table 2 animals-12-01093-t002:** Results of the Lasso regularized multivariate logistic regression models to test the adequacy of behavioral parameters as predictive tools for bovine respiratory disease (BRD) in the last 7 days before diagnosis.

Days Relative to BRD Diagnosis	Test Characteristics					
	AUC ^1^	ACC ^2^ (%)	SE ^3^ (%)	SP ^4^ (%)	PPV ^5^ (%)	NPV ^6^ (%)
−7	0.57	66.7	20.0	93.2	62.5	67.2
−6	0.62	71.4	28.6	95.9	80.0	70.1
−5	0.60	71.7	25.0	95.0	71.4	71.7
−4	0.63	70.3	33.3	92.5	72.7	69.8
−3	0.74	79.0	50.0	97.4	92.3	75.5
−2	0.75	78.0	60.9	88.9	77.8	78.0
−1 −2 and −3	0.57 0.83	67.2 85.7	16.0 71.4	97.6 95.2	80.0 90.9	66.1 83.3

^1^ AUC = area under the curve; ^2^ ACC = accuracy; ^3^ SE = sensitivity; ^4^ SP = specificity; ^5^ PPV = positive predictive value; ^6^ NPV = negative predictive value.

**Table 3 animals-12-01093-t003:** Results of the thoracic ultrasonography in 76 dairy calves. Data for calves classified as diseased with bovine respiratory disease (DIS) and for clinically healthy control calves (CON) are presented.

Ultrasonographic Findings ^1^	Specification	DIS (n = 38)	CON (n = 38)
No abnormalities		7	30
Comet tails	multiple	22	-
	single	-	6
Lung consolidation	≥1 cm	4	-
Pleural effusion		3	-
Lesion	<1 cm	-	2
	≥1 cm	1	-
Abscess	≥1 cm	1	-

^1^ Classification of the ultrasonographic findings as described by Babkine and Blond [24] and Adams and Buczinski [25].

## Data Availability

Data is contained within the article or Appendix A.

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
