# Peer review of "Early Detection of Respiratory Diseases in Calves by Use of an Ear-Attached Accelerometer"

_animals, 2022, doi:10.3390/ani12091093_

Round 1

Reviewer 1 Report

The manuscript animals-1652304 entitled “Early detection of respiratory diseases in calves by use of a 2 ear-attached accelerometer” deals with an interesting topic regarding the use of accelerometers for the early detections of the bovine respiratory disease (BRD). I believe that the use of sensors is becoming more relevant, and their importance in farming will be more significant in the future. I appreciate the manuscript and the research, and congrats to the authors, but concerns about some lacks might be underlined, and more commentaries might be added to clarify some aspects.

As stated by the authors (lines 272 and 333-334), the algorithms used to describe the animal behaviour wasn’t validated for calves. Therefore, we don’t know how this algorithm is affordable for the calves. In general, it would be good to refer to validated algorithms for the five reported behaviours (lying, standing, active, inactive, high active and ruminant. At least can the Authors provide summarizing data about the quality of the used algorithm for the adult bovine (accuracy and other statistics)?

The authors used a case-control study in 41 DIS +41 CON animals. They found that some DIS’s behaviours differ from CON. Are these results applicable also to defined “normal” behaviour found in the 428 clinically healthy calves? In other words, the DIS behaviours statistically differ if compared with the “normal” behaviour?

Limitations of the study might be underlined better. As the authors stated (line 310-311), “normal” behaviour must refer to different age and management conditions (what about race and sex?); therefore, several classes of “normal” behaviours must be described. Moreover, using many other behaviours (such as feeding, standing still, and resting) and postures (such as standing, right sternal recumbency, left sternal recumbency) may be used to improve the predictions. So what do the authors think about this?

Line 156, could the authors specify what “labelled as valid” mean?

Line 165, Is the L1 RMLM calculated over the 41 DIS +CON dataset? Please specify.

Line 168, could the authors specify what “active” or “high active” mean?

Lines 189-191 report some results (i.e. inactive time for d-4, d-2 and d0), but other non reported means are different for DIS and CON (see figure 1). Perhaps a table with all mean values could be written in supplementary material?

Lines 191-193 data refers to daily mean ± standard deviation. Standard deviation looks really high if compared to the mean. For example, data reported in line 240 shows a standard deviation much smaller. Could you clarify? Can you discuss such considerable variability?

Line 191-193, the means for DIS Vs Con differ statistically for day -4 (17.1 Vs 15.8), -2 (18.4 Vs 16.7) and 0 (17.4 Vs 15.4). Compared with the reported standard deviation, such difference looks minimal to me. Are they really appreciable?

Lines 185-199 Results are reported as mean ± standard deviation, and a Shapiro Wilks test for normal distribution have been run. Finally, the authors used a non-parametric test. Could you report if the variables were normally distributed or not?

Figure 1, figure caption does not explain what the bars represent? The 95%CI? Or the standard deviation?

Moreover, if “active” + “inactive” + “high active” = 100, in the day -4, -2, and 0, the authors report differences for “high active” and “inactive” statistically, but not for “active”. Is that correct?

Table 2, Lines 300-303, and conclusion. PPV and NPV were indeed quite adequate, but accuracy and sensitivity were not so high. Could you please discuss better and compare these results with other published research in similar fields?

Lines 321-322 This sentence looks like speculation to me.

Author Response

Dear Reviewer,

we thank you for the time you spent on reviewing our manuscript and all your comments that help to improve our work. Please find our point-by-point answer in the attached file.

Daniela Klein-Jöbstl (corresponding author)

Reviewer 2 Report

Interesting piece of relevant research on application of sensor technologies to dairy calf health.  The authors present a well written manuscript.  The use of cohort analyses is acceptable and relatively common.  I recommend in future analyses of this and similar data the authors collect longer duration data prior to illness.  Doing so would facilitate of using individual animal behavior patterns to detect BRD or other morbidity causes based on use of statistical process control analyses of each individual animal from a healthy baseline.

Author Response

Dear Reviewer,

I would like to thank you vera much for your comments. We will take this into consideration for future research studies.

Best regards

Daniela Klein-Jöbstl (corresponding author)

Reviewer 3 Report

Introduction

Line 41 - reference 2 does not appear to support the statement made. Rather it relates to FPT.

Line 43 - varies

Line 44 - 9.4 is reported for pneumonia by [5] no 9.2 or 22.5% please check quoted figures and the refs align I was unable to access ref [6]. No mention of welfare in [7] do you have another reference to support this aspect of the statement?

Materials and methods

Line 74/75 - …first 2 mo of life… suggestion first two months of life would flow better for the reader.

Line 77 – Although details are given later this first mention of the ear tag ACL is where I would have expected them to appear…

Line 92 – can you briefly expand on this to define ‘high active’, ‘active’ and ‘inactive’ with respect to what this tag is measuring. Also the difference between lying and inactive would be interesting…

Line 96 - do you have a reference for the algorithms developed for the adult cows?

Line 101Subsequently… this is an awkward sentence can you rephrase it to make your point more clearly?

Line 114 – What do you mean by considered for further CEx? How did you decide which of these calves would be examined further? If the calves were not presented for examination, what was the consideration process? I initially thought this was a word error but this step appears in the results as well so please give more detail of this consideration process so your study is repeatable.

(Also in the discussion you mention chasing and head locks – these aspects of animal handling and restraint should appear in the methodology).

Line 115-116 – please define the “F” sound for users unfamiliar with the term.

Table 1 – the legend says adapted from – what were the adaptations made and why? The simple summary implies the scoring was as per ref [1].

Line 136 - randomly implies a process, how was this selection done?

Line 146-147 – results.

Line 151…more in detail… suggestion: in more detail for one day (d X)

Line 164 – …on subsequent days separately … – your intended meaning is not clear here. Can you rephrase to improve the clarity for the first time reader?

Line 165 - Is there a word error here? continuous? Or combined?

Line 170 – just checking, from this I take away that the +1 day data was not treated in this way.

Results

Line 182 – (considering…  suggestion: …and were considered for further CE

Line 187 - was 41

Line 192 - I found the reporting of SD was unhelpful to my understanding here. A SEM or 95% confidence interval would be more useful for the reader in considering the presented data. SD does not tally with what appears in the fig 1 graphics? Consider throughout – although graphics for normal behaviour do display SD and it makes more sense when the axis starts at 0 min/hr.

Figure 1 legend Adjust to indicate what the errors bars denote. Define BRD and add number of calves into legend so the figure stands alone and can be interpreted without reference to the body of the text.

Line 228 - six and two?

Line 253 – daily normal rumination behaviour?

Discussion

Overall I found the discussion lacked detail. Rather than present the evidence for your argument with supporting references, there are generic statements and then listed references. This means the reader needs to go and look up the references to see what evidence was presented, and how your findings relate to the current body of literature.  The discussion would be strengthened by your presenting the actual findings of other workers in the field, and then discussing whether your results agree/ disagree with them.

Line 272 – One limitation of the here used… Awkward sentence,  suggestion:  one limitation of the algorithm used in the current study... What would need to happen for it to be validated for calves? How could not being validated impact your results in the present study?

Line 276 – ‘higher’ how much higher? Can you report the size of the effect, perhaps the range for the studies cited so the reader knows how much higher they are than yours.

Line 277 – ‘high’ criteria… Suggestion: constituted a higher threshold for diseased status...

Line 281- The meaning of this sentence is not immediately clear. Can you rephrase it to make your point clearer. Mixture of single and plural doesn't help understanding.

Line 282 – do refs 27 28 and 29 all speak to all these factors?  More detail outlining how these factors (especially age) could play a role would help the reader understand why your study had limitations.

Line 292 - DIS compared to CON calves.

Lines 296-297 In the literature, results regarding changes in behavioral parameters before onset of disease differ, e.g., for lying behavior [26,33,35,36].

Expand to present your results to the reader in the light of the existing literature. At the moment you are skimming the surface expecting the reader to go and find/read these sources to understand your argument.  You don't make it clear whether you mean results go up/down or whether evidence across studies is in agreement or conflicts with your findings.

Line 303 – expand on the welfare impacts you are talking about.

Line 31316.8 h/day [26,38] two studies cited and only one result presented, suggesting they both reported the same finding (26=16.8, 38=median 15.8, check and include figures). This discussion of age related normal behaviour could be developed more. 

Line 315 - which / what management and environmental factors - what does the literature report?

Line 316 – include handling procedures in methodology, how do you think this could have impacted the results obtained?

Line 318 – “longer” illustrate the size of this effect – then the reader knows how much longer and the statement is more objective. This can be explained by… How? You need to present some evidence to support your argument (some of which has gone before).

Line 320 – nebulous ‘factors’ again, if they had been laid out in detail prior to this statement it would be fine.

Line 325 – word order, the results presented here.

Conclusions

Line 330 - what do you mean by “change” here? How this measure might change with larger sample sized or increased disease pressure...

References

[16] 7th ed.

Author Response

Dear Reviewer,

we would like to thank you for the time you spent for reviewing our manuscript and for all your detailed comments, that help us to improve our manuscript. We hope we could change evreything to your satisfaction.

Please find our answers to each of your comments and questions attached.

Best regards

Daniela Klein-Jöbstl (corresponding author)

Round 2

Reviewer 1 Report

The Authors responded and clarified any doubt or question. The manuscript increased its quality, and I want congrats to them.